# Role of the Cation-Chloride-Cotransporters in Cardiovascular Disease

**DOI:** 10.3390/cells9102293

**Published:** 2020-10-14

**Authors:** Nur Farah Meor Azlan, Jinwei Zhang

**Affiliations:** 1Institute of Biomedical and Clinical Sciences, Medical School, College of Medicine and Health, Hatherly Laboratories, University of Exeter, Exeter EX4 4PS, UK; nm503@exeter.ac.uk; 2Xiamen Cardiovascular Hospital, School of Medicine, Xiamen University, Xiamen 361004, Fujian, China

**Keywords:** cardiovascular disease, hypertension, atherosclerosis, electroneutral transport, cation-chloride-cotransporters, KCCs, NKCCs

## Abstract

The SLC12 family of cation-chloride-cotransporters (CCCs) is comprised of potassium chloride cotransporters (KCCs), which mediate Cl^−^ extrusion and sodium-potassium chloride cotransporters (N[K]CCs), which mediate Cl^−^ loading. The CCCs play vital roles in cell volume regulation and ion homeostasis. The functions of CCCs influence a variety of physiological processes, many of which overlap with the pathophysiology of cardiovascular disease. Although not all of the cotransporters have been linked to Mendelian genetic disorders, recent studies have provided new insights into their functional role in vascular and renal cells in addition to their contribution to cardiovascular diseases. Particularly, an imbalance in potassium levels promotes the pathogenesis of atherosclerosis and disturbances in sodium homeostasis are one of the causes of hypertension. Recent findings suggest hypothalamic signaling as a key signaling pathway in the pathophysiology of hypertension. In this review, we summarize and discuss the role of CCCs in cardiovascular disease with particular emphasis on knowledge gained in recent years on NKCCs and KCCs.

## 1. Cardiovascular Disease

Cardiovascular diseases (CVDs) are the leading cause of mortality globally. It has been estimated that countries in the European region alone spent GBP 80 billion on health care costs related to CVDs [1,2,3]. CVDs are a group of health complications that affect the heart and circulation, including coronary heart disease, heart attack, stroke, heart failure, arrhythmia, and more. Coronary heart disease (CHD) is mainly caused by reduced or blockage of blood flow to the heart and can lead to heart attacks (blocked blood flow to the heart) and heart failure (failure of the heart to pump blood). CHD could also cause heart valve problems and abnormal rhythms (arrhythmias). A stroke occurs due to blockages of blood flow to the brain. Risk factors associated with CVDs include hypertension, diabetes, atherosclerosis, smoking, obesity and high cholesterol. Among the risk factors, hypertension and atherosclerosis are the main underlying causes of CVD. The renal system plays a critical role in blood pressure homeostasis. Water retention in the kidney, due to increased sodium reabsorption, causes extracellular fluid (ECF) volume expansion [4]. The volume expansion contributes to the elevation of preload which contributes to the increase in blood pressure. Hypertension could also lead to atherosclerosis. Atherosclerosis is the process of hardening and narrowing of the blood vessels. Vascular proliferation is one of the contributing factors to the pathophysiology of atherosclerosis. Vascular proliferation is the phenotypic changes of the vascular smooth muscle cells (VSMCs), the most numerous cell type in the blood vessel, characterized by the reversible transition from a quiescent contractile state to a synthetic migratory phenotype [5]. Although there are preventive measures to reduce the burden of CVDs, atherosclerosis is common in people over the age of 65 and 1.3 billion people worldwide are estimated to be hypertensive [1,2,3]. Thus current research aims are to discover novel targets for the treatment of CVDs. Our recent research developed a molecular compound termed ZT-1a for the treatment of stroke, by modulating the activities of cation chloride cotransporters (CCC) [6]. In the review, the focus is to summarise previous work and recent advances that attest to CCC’s growing potential as a therapeutic target for CVDs.

## 2. Evolution of the Cation Chloride Cotransporter Family

Comprehensive phylogenetic analysis by Hartmann et al. revealed the existence of a single ancestral cation chloride cotransporter (CCC) gene in the Archean, *Methanosarcina acetivorans* [7]. The ancestral gene appears to be the base of numerous duplication events which led to paralogous CCC subfamilies in Archean and eukaryotes. These subfamilies consist of sodium chloride cotransporter (NCC), sodium potassium chloride cotransporters (NKCCs, NKCC1 and NKCC2; with NCC collectively referred to as N[K]CCs), potassium chloride cotransporters (KCCs, KCC1–4), polyamine transporter (CCC9) and CCC-interacting protein (CIP1). NCC and NKCCs use Na^+^ in their stoichiometric translocation of Cl^−^, whereas KCCs using K^+^. CIP1 have been shown to inhibit NKCC1 activity [8] and enhance KCC2 [9] activity in cultured cells. CCC9 remains unclassified. CCCs provide electroneutral transport of sodium, potassium and chloride across the plasma membrane; with an exception of NCC, all CCCs are inhibited by several structurally similar compounds, such as bumetanide and furosemide (Table 1). It is important to note that KCCs are only weakly inhibited by these compounds. Further gene-loss events resulted in the complex distribution of CCCs across the taxa.

Gene duplication within the CCC subfamilies of vertebrates resulted in subfunctionalization [7]. This process created a number of isoforms with different expression patterns and functionality in various organs and tissues. Further variants are generated by alternative splicing of the isoforms with various ratios of expression in human tissues (Table 1).

The topology of all CCCs consists of 12 transmembrane domains (TM) domains and large intracellular amino (N) and carboxyl (C) terminal domains [29]. Notably, the KCC and N[K]CC subfamilies vary in their position of the long extracellular loop: between TM5 and TM6 in KCCs and between TM7 and TM8 in the N[K]CC subfamily. Within the isoforms of the subfamilies, the TMs and C-terminus are highly conserved. In contrast, the N terminus is considerably more variable amongst isoforms [29]. Binding of diuretics occurs in the conserved TMs and not the variable termini [29]. Therefore, loop diuretics like furosemide and bumetanide are used to characterize the physiological function of KCCs and NKCCs with little specification for each paralog. The termini are involved in the regulation of transport activity indicative of the sites for post-translational modifications like phosphorylation that are located within them.

Although the family N[K]CC and KCC reciprocally regulate Cl^−^ uptake for N[K]CCs and extrusion for KCCs, the mechanism of coupled ion translocation is considered to be similar. In an alternating access model for NKCCs proposed by Haas, the carriers only move when they are either fully loaded or completely empty [30]. Following ions binding in a strictly ordered sequence, a conformation change occurs that causes the transporter to face the opposite direction and adopt a new form (inwards facing state during extracellular ion binding and vice versa) [30]. The ions are subsequently released in the same order they were bound. Once empty, the transporters return to the original form ready for a new cycle. Recent 3D models have been made that support Haas, however, due to the many differences between paralogs of CCC family members, inferring functional properties from data obtained for a different family member may not be desirable [29].

## 3. Functional Regulation of the Cation Chloride Cotransporter Family

It is generally accepted that regulation of CCC function is accomplished through phosphorylation/dephosphorylation events. Phosphorylation/dephosphorylation reciprocally regulate the N[K]CC and KCC via a network of serine-threonine kinases and phosphatases [31,32,33]. Phosphorylation activates the Na^+^ dependent subfamily and inactivates the K^+^ dependent subfamily and vice versa [34]. The master regulator of CCCs, With-No-K (Lysine) kinases (WNKs) regulate the CCCs via their downstream targets, Ste20/SPS1-related proline/alanine-rich kinase (SPAK) or the SPAK homolog oxidative stress-responsive kinase 1 (OSR1) [31,32,33]. Threonine and serine residues of the CCCs are phosphorylated as a result of this downstream activation. The number and position of phospho-sites vary between different family members as they are not evolutionarily conserved among isoforms and paralogs [29]. Phosphatases such as protein phosphatase 1 (PP1) counterbalance the action of the kinases [35].

Recent data by Lee et al. [36] challenge the current understanding of KCC regulation. Lee and team reported that phosphorylation of KCC2 specific residue, Ser940, by protein kinase C (PKC) has been shown to activate KCC2, an isoform exclusively expressed in the nervous system. However, the phosphorylation of highly conserved threonine residues in all KCCs is ubiquitously associated with inactivation [33,37,38,39]. Thus, the phosphorylation-mediated-activation of KCC2 may be an adaptive mechanism to assist with KCC2 functions in the nervous system [38,40,41]. Gene mutations in WNK1 and WNK4 result in over-activation of the WNK-SPAK/OSR1 pathway, therefore increased phosphorylation and activation of NCC occurs in the kidney, resulting in hypertension [42,43]. Recent studies highlight multiple other signaling factors and regulators of CCC activities that are implicated in cardiovascular disease [44,45,46,47,48]. More specifically, the functional regulation of CCCs in vascular smooth muscle cells and renal cells play a significant role in the pathogenesis of atherosclerosis and Mendelian and non-Mendelian forms of hypertension [45,49,50,51].

## 4. Role of NCC in Cardiovascular Disease

Previous studies have demonstrated that the thiazide-sensitive NCC in the distal convoluted tubule (DCT) plays an important role in blood pressure regulation and that loss of NCC function causes hypotension (Figure 1). Contrastingly, gain-of-function mutations that result in the overactivation of NCC cause Familial Hyperkalemic Hypertension (FHHt) or Gordon’s syndrome, a Mendelian form of hypertension. FHHt is caused by genetic mutations in genes encoding for the regulatory WNK-SPAK/OSR1 and its upstream regulator, an E3 ubiquitin ligase complex containing kelch-like 3 (KLHL3) and Cullin 3 (CUL3). Indeed, current research is focused on identifying novel targets within the WNK-SPAK/OSR1 pathway for use in FHHt and potentially non-Mendelian forms of hypertension. A large body of evidence has demonstrated that inhibition of various components of the WNK-SPAK/OSR1-NCC pathway is an effective drug target for reducing blood pressure [52]. However, the lack of specificity across WNK and SPAK isoforms is challenging for the drug discovery process [45]. A more detailed account of the CUL3/KLHL3-WNK-SPAK/OSR1 pathway as a target for hypertension can be found in the review by Ferdaus et al. [45].

## 5. Role of KCCs in Cardiovascular Disease

Loop diuretics, such as furosemide and bumetanide, are antihypertensive which acts primarily on NKCC2 in the thick ascending limb of the Loop of Henle (TAL) to inhibit NaCl reabsorption and consequently reduce blood pressure [53]. However, loop diuretics have also been shown to act on KCCs [48,54,55]. Although much research has centered on understanding the role of KCC in vascular cells, a growing body of evidence indicates KCC participation in renal physiology [56,57]. Two isoforms, KCC3 and KCC4 are differentially expressed in the kidney reflecting the variability of their role in cardiovascular disease; KCC3 may regulate glucose reabsorption while KCC4 participates in maintaining renal salt and acid-base balance [56,57]. A secondary effect of loop diuretics is vasodilation. Several members of the KCCs (KCC1-3-4) family have been identified in the VSMCs [48,54,55]. This evidence suggests a role for KCC in the VSMCs. Researchers have reported that growth factors such as platelet-derived growth factors (PDGF) [48] that promote and inhibitors such as nitric oxide (NO) [58] that inhibit phenotypic changes of the VSMCs are also modulators of KCCs in the VSMCs. Together with the observation of cardiovascular defects such as arterial hypertension and impairment in cell volume signalling in renal tubules, in three models of KCC3 null mice [59,60], the evidence makes a compelling case for a pathogenic role of KCC beyond its well-studied physiological role in the erythrocytes. Here, we will discuss the substantial body of evidence suggesting a role of KCCs and their signaling pathways in the pathogenesis of cardiovascular disease.

### 5.1. KCC in Vascular Cells

Contractile VSMCs undergo phenotypic changes to proliferative/migratory cells in response to serum factors such as PDGF. Proliferative VSMCs in turn, forms lesions that are evident in vascular diseases such as atherosclerosis. Due to KCC participation in other proliferative diseases such as cancer [61] and reports of the functional significance of cell volume regulation in cell proliferation [62], a role for KCC in cell proliferation and migration is plausible. Indeed, Zhang and colleagues observed inhibition of KCC basal activity following a 24 h serum deprivation and recovery upon serum addition [48]. Using synthetic VSMCs as a model system, an increase in KCC1 and KCC3 activity after short term exposure (10 min) to PDGF and increase in KCC1 but decrease in KCC3 mRNA expression after long term exposure (12 h) was observed [48]. Consistent with evidence that KCC1 and KCC3 mRNA expression follows a 2:1 ratio in VSMCs [63], the results suggest that KCC1 may play a more vital role in PDGF-mediated modulation of VSMCs.

AG-1296, an inhibitor of the PDGF receptor, abolished the PDGF-induced increase in KCC activity, indicating PDGF activation of KCC through its membrane receptors [48]. Dimerization and autophosphorylation of PDGF membrane receptors initiate downstream signal transduction through molecules such as phosphatidyl-inositide-3-kinases (PI3-K) and mitogen-activated protein kinase (MAPK). Pharmacological inhibition demonstrated that PDGF-mediated activation of KCC occurs via the P13K/Akt pathway [64]. Although the inhibition of MAPK by PD98059 has been shown to suppress KCC activity in red blood cells [65] and breast cancer cells [66], no evidence of MAPK involvement in PDGF activation by KCC was found. Notably, the study on breast cancer cells found that stimulation of KCC4 mRNA levels by insulin-like growth factor-1 (IGF-1), another serum growth factor, was inhibited by 65% with an inhibitor of ERK1/2, PD98059, but was insensitive to the p38 MAPK inhibitor, SB202190 [66]. While inhibitor of Akt, LY 294002, only inhibited IGF-1 stimulation of KCC4 mRNA levels by 35% [66], suggesting a larger role for ERK1/2 in the regulation of KCC expression. However, it is important to note that the length of incubation varied between the experiments: 1 h for VSMCs and 3 h for red blood cells and breast cancer cells. Therefore, the importance of each signaling pathways could vary between cell types. The researchers also found that calyculin A, an inhibitor of protein phosphatase 1 (PP1), significantly inhibited PDGF regulation of KCC by 60% [64], suggesting the participation of PP1 in KCC modulation by the PDGF-PI3K pathway (Figure 2). As there is no clear evidence of PI3K regulation of PP1, further research will need to determine whether additional components of downstream of PI3K exists is crucial.

Although the evidence is clear that PDGF regulates KCC, it is not clear how KCC activation plays a role in PDGF-mediated signaling to promote atherosclerosis. Recent kinetic studies of KCC activity found evidence to support direct KCC involvement in vascular phenotypic changes [50]. The researchers found significant changes in the kinetic parameters for the co-transport of Cl^−^ between early, intermediate and late VSMC synthetic phenotypes. Increased KCC1 and KCC4 expression in the late synthetic VSMC in comparison to earlier states suggest a functional role of KCC in phenotyping switching to a diseased state.

Nitric oxide acts to regulate the function of VSMCs via cGMP-dependent activation of protein kinase G (PKG) to promote VSMC relaxation. Activators of the NO/cGMP/PKG pathway such as sodium nitroprusside, NONOates (NO donors) and 8Br-cGMP (PKG substrate) have been shown to stimulate KCC activity, mRNA and protein expression. Inhibitors such as KT5823 (PKG inhibitor), calyculin A and genistein (PP1 inhibitor) inhibit KCC activity, mRNA and protein expression [67]. Researchers also found that when the vasodilators hydralazine and sodium nitroprusside were added to pre-contracted arteries, the arteries relax despite the blocking of all pathways except for KCC [67]. Interestingly, the NO signaling preferentially increased KCC3b mRNA 8.1-fold in comparison to 2.5-fold for KCC3a isoform despite a 3:1 KCC3a:KCC3b mRNA expression [68]. Overall, this indicates KCC contribution to the vascular effects of these vasodilators. Studies by Di Fulvio and team reported NO/cGMP/PKG signaling pathway and PKG participation in the regulation of KCC1 [58] and KCC3 [63] mRNA expression respectively. Further studies by Adragna et al. found that baseline KCC activity was higher in PKG transfected cells (PKG+) in comparison to PKG deficient cells (PKG-) [55]. Furthermore, the deletion of the KCC3 isoform has been shown to cause hypertension in mice [60,69]. These findings suggest that KCC may be involved in vasodilation through the NO/cGMP/PKG-PP1 pathway (Figure 2).

A common modulator of both the PI3K/AKT and the NO/cGMP/PKG signaling cascades in the VSMCs is Apelin [70]. Apelin stimulates the NO pathway to induce vasodilation and the PI3K/Akt pathway to induce proliferation and migration of VSMCs [71]. Due to its cardioprotective effects in the VSMC’s, Apelin has been suggested as a novel therapeutic target in the cardiovascular system [71]. In 2013, researchers found that Apelin stimulated KCC activity through the NO pathway by 336% and 142% by the MAPK/PI3K pathway [72]. In contrast, oxidized plasma cholesterol (oxLDL), a known promoter of vascular lesions, inhibited KCC by 70% and treatment with Apelin restored this function [72]. These studies prove that KCC3 has a physiological role in the vascular cells and thus has potential as a therapeutic target, however it is not clear if KCC acts through modifying the membrane potential, changes in cell volume, or influence of intracellular chloride activity.

A study by Rust et al. reported an increase in [Cl^−^]_i_ but no difference in vascular contractility in KCC3 knockout mice [69]. Instead, pharmacological inhibition of the sympathetic nervous system reduces blood pressure after 80 s and produced an increase in urinary excretion of catecholamine. Additionally, the isolated arteries from the KCC3 null mice did not respond differently to vasoactive interventions in comparison to wild type (WT) mice [69]. This suggests that KCC3 inactivation likely contributes to hypertension through neurogenic mechanisms and that the increase in [Cl^−^]_i_ did not significantly affect vascular contractility. However, recent studies by Garneau et al. found normal levels of circulating catecholamine, supporting a role for KCC3 in the vasculature [73]. It is important to note that although both groups studied contractile properties, the study by Rust used isolated saphenous arteries while Garneau et al. used thoracic aortas. Furthermore, the isolated saphenous arteries used by Rust were hypertrophied and presented with increased [Cl^−^]_i_ compared to the WT littermates. Thus more than one mechanism could be responsible for the cardiovascular phenotype.

Taken together, these studies identified KCC as a key mediator of vascular pathologies. KCCs are implicated in atherosclerosis through their regulation by PDGF and in hypertension via regulatory NO (Figure 2). It is important to note that although the functional role of each KCC isoform remains obscure due to the lack of specific inhibitors for each KCC isoform, the comprehensive effects of KCC on VSMC proliferation is evident.

### 5.2. KCC in Renal Cells

Three isoforms, KCC1, KCC3 and KCC4 were found to be expressed in the kidney [74,75]. Although KCC1 is ubiquitous, its expression has only been studied at the mRNA level [22], thus the role of KCC1 in the kidney remains unknown. KCC3 is exclusively expressed in the basolateral membrane in the proximal tubule [76] and KCC4 is expressed in the basolateral membrane of proximal tubules [77], TAL and in alpha intercalated cells in the collecting duct. Although the first KCC isoform, KCC1, was identified in 1996 [22], in 1992, KCC was shown to be elevated in patients suffering from Liddle syndrome, an inherited form of high blood pressure [78], suggesting a tubular origin of the disease. However, the authors could not find further studies regarding the pathogenic role of KCC in Liddle syndrome. The primary defect in Liddle syndrome has since been identified to be mutations in the epithelial sodium channels (ENaC) subunits [79,80]. Melo et al. evaluated the expression level and distribution of KCC3 and KCC4 in rats exposed to hyperglycaemia, a low-salt diet, metabolic acidosis and low or high K^+^ diet [56]. Consistent with the previous discovery of glucose stimulation of basolateral efflux of K^+^ ions [81], Melo et al. found that KCC3 mRNA and protein expression were increased during hyperglycaemia but not with low-salt diet or acidosis. In contrast, KCC4 protein expression was increased 1.53 ± 0.16-fold by a low sodium diet in the kidney and metabolic acidosis specifically in the alpha-intercalated cells of the collecting duct (CD) [56]. Overall, the results suggest that KCC3 is only involved in the glucose reabsorption mechanism as there is no overt renal phenotype detected in the KCC3 knockout mice models. Although the mice display arterial hypertension, the exclusive expression of KCC3 in the proximal tubule disavow their contribution to the hypertensive defects observed. Moreover, a causal relation between hypoglycemia to cardiovascular events has been much debated, thus, further research is needed. Contrastingly, the upregulated KCC4 expression in low-salt diet and metabolic acidosis implicates KCC4 in salt reabsorption of the TAL and acid secretion in the CD. More specifically, the increase in KCC4 activity during salt restriction could be a compensatory mechanism to promote basolateral K^+^ efflux to support salt transport. The findings from Melo are consistent with previous studies by Boettger et al. who reported renal tubular acidosis in KCC4 deficient mice [27]. Boettger reported compensated metabolic acidosis along with increased urine alkalinity in KCC4 knockout mice (pH 7.3 ± 0.1) in comparison to the WT littermates (pH 6.4 ± 0.1) [27]. Energy-dispersive X-ray microanalysis revealed increased [Cl^−^]_i_ in alpha-intercalated cells of KCC4 knockout mice. The alpha-intercalated cells in the kidney secrete acid via the apical H-ATPase and H^+^/K^+^ exchanger and reabsorb bicarbonate via the basolateral Cl^−^/HCO_3_^−^ exchanger (Figure 1) [57]. The rise in [Cl^−^]_i_ indicates a more alkaline intracellular pH and electrochemical imbalance which inhibits apical H^+^ secretion, resulting in type 1 renal tubular acidosis. Cl^−^ extrusion through KCC4 is important to negate the influx through the Cl^−^/HCO_3_^−^ exchanger. Cl^−^ import and bicarbonate extrusion through the Cl^−^/HCO_3_^−^ compensate for the acid secretion on the apical side. Thus, KCC4 plays a physiological role in renal acidification. Loss of KCC4 leads to renal tubular acidosis, which presents with hypokalemia [57]. If hypokalemia is left untreated, cardiac arrhythmias can occur. In addition, renal tubular acidosis contributes to overall metabolic acidosis, an imbalance in the acid-base homeostasis. Metabolic acidosis commonly occurs in patients with chronic kidney disease (CKD). Although the relations between metabolic acidosis and CVD are obscure and thus require further studies, CVD is the leading cause of mortality in patients with CKD.

## 6. Role of NKCC in Cardiovascular Disease

Loop diuretics such as furosemide and bumetanide are antihypertensive, inhibiting sodium reabsorption in the kidney and consequently reducing blood pressure [53]. Loop diuretics are the mainstay treatment to promote diuresis in patients with congestive heart failure. Loop diuretics bind to the TM of NKCCs, a region that is highly conserved, and thus are not specific to one subfamily or their isoforms [29]. As such, pharmacological studies have revealed key roles for the ubiquitous NKCC1 and renal specific NKCC2 in vascular and renal cells respectively. NKCC1 is involved in the suppression of myogenic response in microcirculatory beds [51,82] and NKCC2 is involved in blood pressure regulation through sodium reabsorption.

### 6.1. NKCC in Vascular Cells

Studies by Garg et al. found that bumetanide is less efficient in hypertension due to a fixed aortic coarctation, a model for a fixed increase in resistance in the aorta, in comparison to hypertension by continuous infusion of norepinephrine [83]. One limitation of the study is that BP was measured under anesthesia. Despite the limitation of BP measurement under anesthesia, and the ambiguity around blood pressure regulation of NKCC1 or vice versa, it can be concluded that the hypotensive action of bumetanide is through systemic vascular resistance rather than cardiac output and that NKCC1 influences blood pressure through the smooth muscle tone in resistance vessels [84]. Like all muscle cells, VSMC responds to the rise in intracellular [Ca^2+^] concentration ([Ca^2+^]_i_) as a trigger for contraction. Thus myogenic tone is dependent on the influx of calcium [Ca^2+^] into VSMCs via voltage-dependent Ca^2+^ channels and intracellular stores. Cl^−^ channel activation leads to Cl^−^ efflux, which depolarizes the membrane. This activates the voltage-dependent Ca^2+^ channel and elevates [Ca^2+^]_i_. The rise in [Ca^2+^]_i_ mediates myosin-actin interaction, cross-bridge cycling and consequently, VSMC contraction [85]. In the VSMC, NKCC1 is responsible for the inward-directed flux of Cl^−^. Consistent with the observation of reduced blood pressure in NKCC1 null mice, inhibition of NKCC1 by bumetanide decreased [Cl^−^], reduced Ca^2+^ uptake and completely blocked contraction [86]. This observation coupled with reports of lack of effect by bumetanide on contraction in response to KCl, which depolarizes smooth muscle cells directly, suggests that NKCC1 contributes to vascular tone via maintenance of [Cl^−^]_i_. Further studies have provred that this is accomplished through the channel opening of NKCC1 which increases the [Cl^−^]_i_ above the electrochemical equilibrium which depolarizes the VSMC through activation of voltage-gated Ca^2+^ [87]. Consistent with previous findings, more recent studies in intact isolated thoracic aortas demonstrated that NKCC1 is involved in phenylephrine-induced rhythmic contraction in the mouse aorta and that NKCC1 is regulated by calcium sparks [88].

The myogenic tone is the ability of blood vessels to constrict. Constriction of vessels, elevates peripheral resistance, consequently raising blood pressure. Pharmacological inhibition with bumetanide, a potent inhibitor of both isoforms of NKCC, was found to suppress and block myogenic tone in mouse mesenteric arteries and rat afferent arterioles respectively [89,90]. However, the inhibitory action of bumetanide on myogenic tone and contractions are absent in NKCC1 null mice. This indicates that high ceiling diuretics (HCDs) influence myogenic tone via NKCC1 and that NKCC1 inhibition could lower blood pressure through effects on vascular cells. This is further supported by Meyer et al. who reported that tail-cuff measurements of blood pressure in NKCC1 deficient mice were significantly lower than the WT mice [84,91]. Although previous studies using the same measurement method found no statically significant reduction in blood pressure in NKCC1 null mice [92], the larger number of mice used by Meyer et al. and the longer length of the time period (21 days) suggests a role for NKCC1 in the maintenance of vascular tone. However, the BP measurements for the WT mice are quite high. This could be due to the tail-cuff method employed by the group as the method is operator dependent thus may produce less reliable results. Studies using radio telemetry, a method widely recognized as the best for establishing blood pressure phenotype, did not find a difference in NKCC1 KO mice vs. WT mice but found a difference when dietary salt was changed [84,93]. Despite the promising in vitro data from isolated vessels, studies of the whole animal are limited due to compensatory changes by the kidney. It is important to consider that although mice lacking NKCC1 have reduced blood pressure, the contractile effect of NKCC1 does not necessarily translate into the effects of blood pressure in vivo.

While NKCC1 is implicated in the maintenance of vascular tone, it may itself be regulated by blood pressure. A study by O’Donnell et al. showed that NKCC1 activity of the VSMC from spontaneously hypertensive rats is significantly less in comparison to normotensive rats [94]. Evidence also showed that NKCC1 activity is upregulated in different models of hypertension and is accompanied by an increase in [Cl^−^]_i_ [95]. These models include hypertension elicited by deoxycorticosterone (DOCA)/salt and angiotensin 2 they showed that these caused increased levels of NKCC1 mRNA. Furthermore, aldosterone increases NKCC1 activity. Although it is clear that NKCC1 is implicated in the maintenance of vascular tone, NKCC1 upregulation in hypertensive models raises the question of whether NKCC1 activity and upregulation increase blood pressure or are the result of hypertension. Using rat aorta, Jiang et al. showed that after aortic coarctation, a procedure that yields both hypertensive and hypotensive segments of the aorta in the same animal, the NKCC1 activity of the hypertensive segment of the aorta is 62% greater than the control. There was also a 21% decrease in NKCC1 activity in the hypotensive segment of the aorta [96]. This is accompanied by observations of a fivefold increase in NKCC1 mRNA in the hypertensive aorta compared with the hypotensive or normotensive aorta. This supports the theory that blood pressure regulates NKCC1 in VSMCs.

There is also evidence that epigenetic regulations of NKCC1 may play a role in its effects on the vascular tone [97]. A study by Lee et al. found that the expression of NKCC1 mRNA and protein levels in the aorta and heart tissues are higher in spontaneous hypertensive rats (SHR) than in normotensive rats [51]. Lee also found greater hypomethylation of the NKCC1 gene promoter in the aorta and heart of the SHR relative to the normotensive rats. This was accompanied by increased NKCC1 expression and inhibitory action of bumetanide on mesenteric artery contractions with age in SHR but not wild type. The Emax for the dose–response curve was 74 ± 2.3 for SHR and 102 ± 4.7 for normotensive rats. This suggests that promoter hypomethylation upregulates NKCC1 in SHR. Lee concluded that the upregulation of NKCC1 could be responsible for the development of high blood pressure in SHR. NKCC1 promoter hypomethylation could therefore be a marker of hypertension development. Further research by the same group found that the expression of NKCC1 is epigenetically regulated during postnatal development of hypertension [98]. Sequencing revealed that the NKCC1 promoter in wild type rats was increasingly methylated with age (8.5%) but remained largely hypomethylated in SHR (2.2%) during postnatal development of hypertension. The activity of DNA methyl transferase 3B (DNMT3B), a family of enzymes that transfer a methyl group to promoter regions to downregulate their expression, is also three-fold higher in the aorta of wild type rats in comparison to SHR at 18 weeks of age. This suggests that the maintenance of hypomethylation of the NKCC1 promoter as a result of decreased DNMTB3B activity is the reason for the age-dependent development of hypertension in SHR. In a further study, Lee et al. noted that inhibition of DNMT with an inhibitor in the rat cerebral cortex resulted in the upregulation of the transcription of NKCC1 during postnatal maturation [46]. Another research group reported that DNMT inhibition also resulted in an increase in blood pressure [99]. Contrary to previous findings, these results could infer that age and blood pressure alter the NKCC1 promoter epigenetically. To elucidate the relationship between blood pressure and NKCC1 methylation, further study on vascular cells is required.

NKCC1 can additionally be upregulated via histone modification in the aortas of angiotensin II (ang-II) induced hypertensive rats. Cho et al. found that the level of NKCC1 mRNA and protein in the aortas increased gradually in ang-II infused rats [100]. A histone activating code, acetylated histone H3, was increased and histone repressive code, trimethylated histone H3, was reduced in ang-II infused rats compared to the sham infused animals. Studies have shown that the recruitment of specificity protein 1 by CpG hypomethylation leads to the upregulation of NKCC1 in hypertensive rats [82]. This could mean that NKCC1 is epigenetically upregulated by histone modification or DNA demethylation upon the development of hypertension. The methylation of CpG dinucleotides may play a role in the epigenetic maintenance of blood pressure through decreased expression of NKCC1. These findings further emphasize NKCC1 methylation or histone modification as a potential biomarker in the diagnosis and management of hypertension. More recently, studies by Ji [101] and Andersen [102] et al. found increased NKCC1 activity after cardioplegia induced arrest of diabetic hearts and in post-infarction heart failure, respectively. These findings indicate the potential detrimental role of enhanced NKCC1 in the diabetic heart after cardioplegia infusions and in the promotion of adverse remodeling after myocardial infarction.

These studies collectively suggest that inhibition of NKCC1, specifically in the smooth muscle could be a pharmacological target for hypertension. However, the ubiquitous nature of NKCC1 coupled with the possibility of concurrent inhibition of NKCC2 in TAL, would preclude currently available NKCC1 inhibitors to be used in the treatment of cardiovascular disease. This is due to the large doses required to achieve an inhibitory response in the plasma which would produce unacceptable toxicity elsewhere [103]. Thus, a compound that selectively inhibits NKCC1 over NKCC2 and is not excreted via the urine is desirable. Indeed, Savardi et al. have now discovered a new drug candidate, termed ARN23746, which selectively inhibits NKCC1 versus NKCC2 in vivo [16].

### 6.2. NKCC in Renal Cells

Despite the kidney-specific expression of NKCC2, NKCC1 has been found in several locations in the kidney including the inner medullary collecting duct. The basolateral expression of NKCC1 results in salt and water secretion whereas the apical expression of NKCC2 in the thick ascending limb of the Loop of Henle results in salt reabsorption (Figure 1). Although mice that lack NKCC1 expression show impaired epithelial chloride secretion in the gut and trachea, later studies have not produced evidence to support a significant role of NKCC1 in chloride secretion in the kidney [104]. Loss-of-function mutations of NKCC2 results in the salt-wasting phenotype of type 1 Bartter Syndrome [105]. The secretory NKCC1 is also upregulated in metabolic acidosis [106]. An extensive account of the role of NKCC1 and NKCC2 in hypertension can be found in a recent review by Orlov et al. [107].

## 7. Hypothalamic Signaling Mechanisms in Hypertension

The hypothalamus is crucial for the central control of blood pressure in response to central and peripheral stimuli. As such, there have been many studies implicating hypothalamic signaling mechanisms in hypertension. For the purpose of this review, we will only focus on the signals that directly involve CCCs. Those are the hypothalamic neuronal signaling mechanisms and the hypothalamic regulation of vasopressin secretion. A complete review of hypothalamic signaling mechanisms in hypertension can be found in a review by Carmichael et al. [49].

Accumulating evidence implicates increased sympathetic drive from the paraventricular nucleus (PVN) in the development of hypertension. More specifically, the hyperactivity of the PVN contributes to the elevated sympathetic drive. The excitability of the PVN is regulated by excitatory glutamatergic and inhibitory GABAergic inputs. Although GABA is the main inhibitory neurotransmitter in the brain, there is evidence that GABA can also elicit an excitatory response. The polarity of the neurotransmitter depends on the GABA equilibrium potential (E_GABA_). A more positive E_GABA_ results in excitatory signaling and a more negative E_GABA_ results in inhibitory signaling. The E_GABA_ is influenced by [Cl^−^]_i_. If the [Cl^−^]_i_ is high, the E_GABA_ will be more positive and the GABA signaling will result in neuronal depolarization. If [Cl^−^]_i_ is low, E_GABA_ will be more negative and the response is an inhibitory and result in hyperpolarization. The [Cl^−^]in neurons are determined by two cation-chloride cotransporters, NKCC1 and KCC2 [108]. Thus, disruption of Cl^−^ homeostasis leads to an imbalance of inhibitory and excitatory signals in the PVN.

In SHR, GABAergic inhibition in the PVN is impaired [109]. Studies by Ye et al. revealed that the E_GABA_ undergoes a depolarizing shift in SHR [47]. The inhibition of NKCC1 normalizes the E_GABA_ and restores GABA inhibition of PVN in SHRs [47]. The mRNA and protein levels of NKCC1 but not KCC2 are significantly increased in SHRs and the inhibition of NKCC1 significantly reduces the sympathetic vasomotor tone [47]. This study provides evidence that NKCC1 activity is important for chloride homeostasis in the PVN. A disruption of the homeostasis diminishes the inhibitory effects of GABA resulting in the increased sympathetic outflow observed in hypertension. The hyperactivation of the PVN is supported by studies in a genetic mouse model of hypertension [109,110,111].

Arginine vasopressin (AVP) is secreted by magnocellular neurosecretory cells (MNC) in the PVN. AVP stimulates water reabsorption in the kidney and causes vasoconstriction to maintain plasma osmolality and blood pressure respectively. In the development of salt-sensitive hypertension (DOCA), AVP secretion is dysregulated resulting in impairment in renal ionic absorption [112]. Studies by Yi et al. found an increase in AVP expression in the AVP of SHR [113]. This change increases both with age and magnitude of hypertension and is due to reduced GABAergic input from baroreceptors in response to high salt intake. Further findings by Kim et al. found that the inhibitory to excitatory switch of GABA receptors in AVP neurons contributes to the increased release of AVP observed in hypertension [112]. The switch is driven by the upregulation of NKCC1 and downregulation of KCC2. Recently, findings suggest that the downregulation of KCC is mediated by brain-derived neurotrophic factor via the tyrosine kinase receptor B (TrkB) pathway [44]. The downregulation of KCC prevents the inhibitory GABAergic signaling evoked by the baroreceptor which leads to increased excitability of AVP secreting neurons [44]. Although these studies confirm the significance of chloride homeostasis in the control of blood pressure, the role of vasopressin in neuronal mechanisms in hypertension still needs to be further elucidated.

Viewed collectively, these studies provide insight into the regulation of sympathetic activity and chloride homeostasis in normotensive and hypertensive phenotypes in vivo. Sympathetic activity is increased in hypertensive phenotypes due to disruption in chloride homeostasis following the upregulation of NKCC1. AVP secretion is increased due to the downregulation of KCC2. This suggests that further investigation of these mechanisms may uncover multiple therapeutic targets to reduce sympathetic activity and AVP release in hypertension.

## 8. Role of Regulatory WNK-SPAK/OSR1 Pathway in Cardiovascular Disease

Mutations in WNK1 and WNK4 genes caused a Mendelian form of hypertension, Familial Hyperkalemic Hypertension (FHHt) also known as pseudohypoaldosteronism type 2 or Gordon’s syndrome [42]. Although WNK1 is expressed in the mammalian DCT, expression levels are low and the predominant isoform in the DCT is a kidney-specific short isoform of WNK1 (ks-WNK1). Disease-causing mutations in the WNK1 genes are intronic deletions that lead to the ectopic expression of full-length WNK1 and ks-WNK1 [114]. Mouse models overexpressing WNK1 display enhanced phosphorylation of NCC [115]. Further research identified a missense mutation in WNK4 (D564A) as the cause of hypertension in FHHt via activation of the WNK-OSR1/SPAK-NCC cascade [42]. This is supported by the observation that SPAK deficiency rescues FHHt caused by WNK4 mutation [116]. Although WNK4 is generally accepted as an activator of NCC, studies have found that WNK4 modulates NKCC2 in vivo [117]. Knockout mouse models of WNK4 have been observed to possess lower levels of phosphorylated NCC and NKCC2 [84]. Homozygous WNK1 knockout in mice is embryonic lethal, however, studies of WNK1 heterozygous mice revealed a significant decrease in blood pressure [118]. Conversely, a study by Susa et al. found that the blood pressure in WNK1 heterozygous mice was not reduced even when fed a low salt diet. The team also did not find a significant decrease in the phosphorylation of OSR1, SPAK, NCC, NKCC1 and NKCC2 in the kidney [119]. In contrast, a significant decrease in the phosphorylation of NKCC1 in the aorta and decreased pressure-induced myogenic response in the mesenteric arteries was observed in WNK1 heterozygous knockout mice. This is further supported by studies completed by the Bergaya team [120] who did not find a decrease in basal systolic blood pressure despite the use of radiotelemetry. Consistent with Susa et al., Bergaya et al. also found a major loss of contractile myogenic response in WNK1 heterozygous mice that are associated with decreased phosphorylation level of WNK1 substrate SPAK and its target NKCC1 in arteries. These studies confirm the contribution of NKCC1 to the regulation of blood pressure and suggest a role for WNK1 in the regulation of NKCC1. Further studies in WNK1 knockout mice revealed that WNK1 is required for angiogenesis and heart development [121]. SPAK knockout mice present with Gitelman syndrome (hypokalemic metabolic alkalosis, hypomagnesemia and hypocalcaemia) and impaired vasoconstriction [122]. Thus SPAK has been suggested as an important pharmacological target for the treatment of essential hypertension [123]. More recent studies have shown that the upstream regulator of WNK, KLHL3 and CUL3, are implicated in FHHt. In particular, mutations in CUL3 causes severe hypertension by affecting both renal and vascular function [124,125,126]. More specifically, mice with mutant CUL3 protein showed increased expression of RhoA, a molecule involved in the regulation of vascular tone [124]. Further information on the mechanisms of CUL3/KLHL3 pathogenesis in FHHt can be found in a review by Ferdaus et al. [127].

We employed studies of phosphoproteomics and functional kinomics and found that WNKs regulate SPAK/OSR1, facilitating the phosphorylation of KCCs. This consequently reduces KCC activity, and the dephosphorylation of NKCC1, increasing NKCC1 function [33,39]. Thus, the same kinase pathway produces inverse effects on the opposing co-transporters, providing a powerful push–pull regulatory control of [Cl^−^]_i_. Through the use of an in vivo SPAK mouse model, we then uncovered a role for SPAK/OSR1 as a bridge to facilitate the signaling cascade between WNKs and CCCs [128]. SPAK/OSR1 was previously shown to phosphorylate CCCs earlier by the Delpire and Forbush labs [129,130]. Furthermore, we have developed a novel SPAK binding inhibitor, termed ZT-1a, which specifically blocks the WNK-SPAK/OSR1-CCC signaling pathway, subsequently reducing the NKCC1 and KCCs phosphorylation in cultured cells, and in vivo mouse and rat tissues [6]. Although further studies are needed, this is promising for the treatment of cardiovascular disease as ZT-1a is an effective SPAK modulator.

## 9. Conclusions

Together, these reviews highlight the potential role of CCCs in cardiovascular disease. The gain of function mutations of WNKs, resulting in the enhanced activity of NCC in the kidney, is the cause of FHHt. An increase in KCC activity in vascular cells could lead to atherosclerosis and have potential vasodilation effects. Impairment of KCCs in renal cells contribute to renal acidosis, a disease which, if not treated, could lead to cardiac arrhythmias. NKCC1 is epigenetically upregulated in hypertension. Impairment of NKCC2 in renal cells can cause Bartter syndrome. CCCs are also involved in hypothalamic signaling in hypertension and the mutations in the regulatory CUL3/KLHL3-WNK-SPAK/OSR1 pathway contribute to the pathology of FHHt. The future development of interventional strategies should exploit these recent findings pertaining to the role of CCCs in vascular and renal cells as well as the pathophysiological mechanisms involved in cardiovascular disease.

### Box 1 Measuring KCC3 and NKCC1 Activity Ion-Transport Activity

The cation chloride cotransporters (CCC) are electroneutral thus electrophysiology methods that use voltage changes and electrical current cannot be used to measure CCC activity. Instead, researchers exploit the characteristic difference of the intracellular concentration of the ions to measure their activity instead. The most commonly used method is unidirectional Rb^+^ flux analysis to measure cotransporter activity [131]. The method uses non-radioactive Rubidium (^86^Rb^+^) as a potassium congener and measures the flux of ^86^Rb^+^. In order to characterize the transporter itself, all other transporters are blocked. For example, when studying KCC3 activity in VSMCs, ouabain and bumetanide were used to block Na^+^/K^+^ pump and NKCC1 activities. This method was used to clone and characterize the function of NKCCs [132] and the function of KCC3 [133]. More importantly, the technique allows for the study of channel properties in a variety of conditions including homogenous cell preparations; erythrocytes, oocytes and different heterologous cell lines. However, it is difficult to employ the methodology in brain slices and neuronal culture. Alternatively, fluorescent assays to measure the movement of thallium ions through potassium channels can be used [84]. This method uses TI^+^ as a surrogate of K^+^ and a TI^+^ sensitive fluorescent dye (FlixORTM) to visualize TI^+^ uptake through channels in single cells. The Delpire group reported similar results of both ^86^Rb^+^ and FluxOR methods [134]. Other techniques include the ammonium pulse technique which uses cardiac strips to measure the acidification rate [135]. A comparative analysis of measuring cation-chloride-cotransporter activity can be found in a review by Medina et al. [131].

## Figures and Tables

**Figure 1 cells-09-02293-f001:**
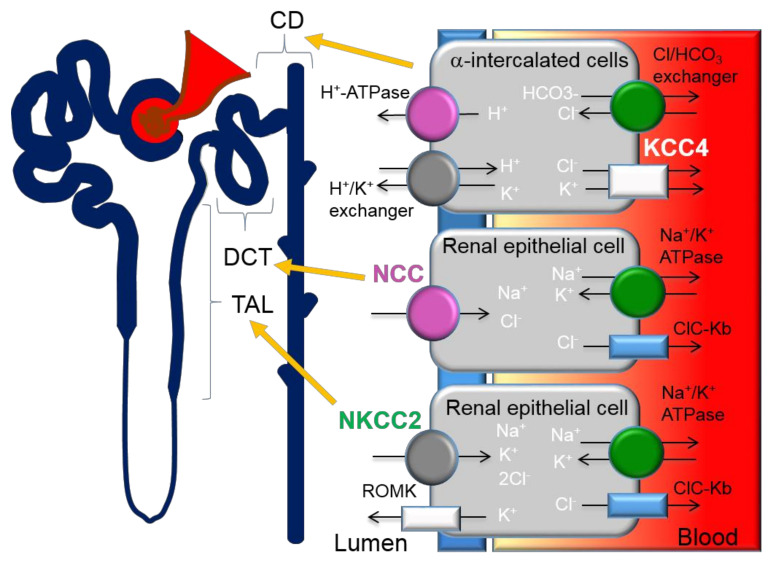
Expression of cation-chloride-cotransporters in the kidney. Alpha-intercalated cells in the collecting duct (CD) secretes acid via the apical H-ATPase and H^+^/K^+^ exchanger and reabsorbs bicarbonate via the basolateral Cl/HCO_3_ exchanger. Efflux of Cl^−^ through the potassium chloride co-transporter-4 (KCC4) is important to maintain the electrochemical gradient to facilitate the acid secretion activities of the alpha-intercalated cells. Loss of KCC4 leads to renal tubular acidosis, which, if left untreated, could lead to cardiac arrhythmias. Sodium-chloride-cotransporters (NCCs) are exclusively expressed in the distal convoluted tubule (DCT). Gain-of-function mutations in regulatory genes that lead to the over activation of NCC cause Familial Hyperkalemic Hypertension (FHHt). Sodium potassium chloride cotransporters-2 (NKCC2) is expressed in the thick ascending limb (TAL). Loss-of-function mutations of NKCC2 results in the salt-wasting phenotype of type 1 Bartter Syndrome.

**Figure 2 cells-09-02293-f002:**
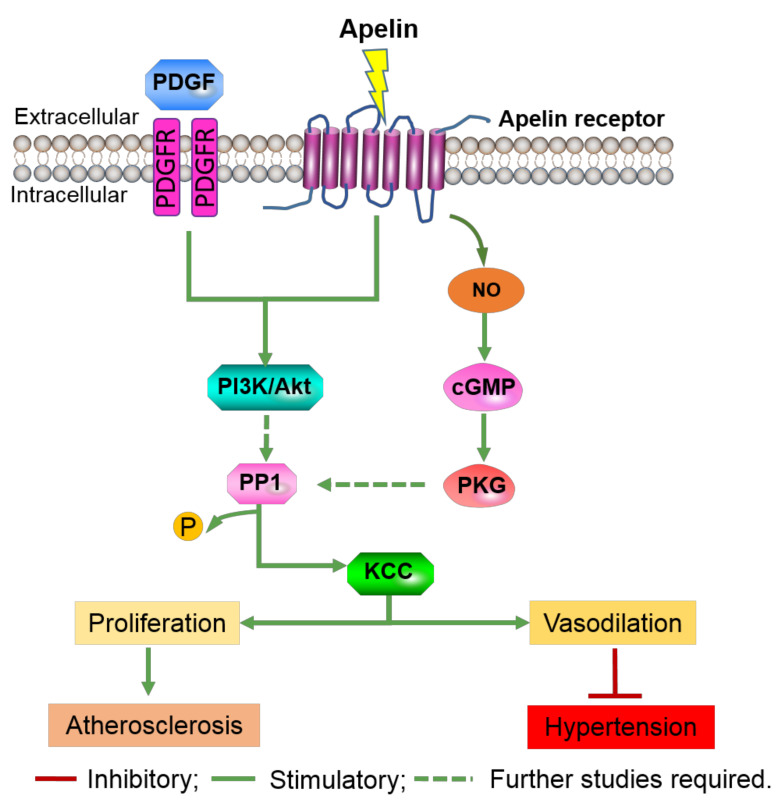
KCC is implicated in atherosclerosis and is a potential therapeutic target for hypertension. Potassium chloride channel (KCC) in vascular cells is regulated via the platelet-derived growth factor (PDGF) and nitric oxide (NO), both of which are implicated in atherosclerosis and vasodilation respectively. PDGF regulates KCC through the dimerization of PDGF receptors (PDGFR) and the subsequent activation of phosphoinositide 3-kinase (PI3K). Through an unknown mechanism, PI3K activates protein phosphatase 1 (PP1) which dephosphorylates and actives KCC. Activation of KCC could enhance the phenotypic switching of vascular smooth muscle cells (VSMCs) into a diseased state. NO regulates KCC via the nitric oxide/cyclic guanosine monophosphate/protein kinase G (NO/cGMP/PKG) pathway. PKG activates PP1 through an unknown mechanism leading to KCC activation and consequent vasodilation. KCC could be a potential therapeutic target for hypertension. Apelin is cardioprotective and a common modulator of both pathways.

**Table 1 cells-09-02293-t001:** Major characteristics of Cl^−^-coupled cation cotransporters [10,11,12,13].

Gene	Human Chromosome Localization	Protein	Transported Ions	Alternative Spicing	Tissue Distribution, Cellular/Subcellular Expression	Link to Disease	Inhibitors, IC50 (μM)
SLC12A2	5q23.3	NKCC1	Na^+^, K^+^, Cl^−^	Isoforms A and B [14]	Ubiquitous: basolateral membrane of epithelial cells, non-epithelial cells	Schizophrenia [15]	Bumetanide, 0.05–0.60; Furosemide, 10–50; ARN23746 [16] Azosemide, 0.246–0.197
SLC12A1	15q21.1	NKCC2	Na^+^, K^+^, Cl^−^	Isoforms A, B and F [17]	Kidney-specific: apical membrane of the thick ascending limb	Bartter syndrome type I [18]	Bumetanide, 0.10–0.50; Furosemide, 15–60
SLC12A3	16q13	NCC	Na^+^, Cl^−^	NA	Kidney-specific: apical membrane of the distal convoluted tubule	Gitelman syndrome [19]	Hydrochlorothiazide, 70 [20]; Metolazone, 0.3 [21]
SLC12A4	16q22	KCC1	K^+^, Cl^−^	NA	Ubiquitous	NA	Bumetanide, 60 [22]; Furosemide, 40 [22]; DIOA, ~10
SLC12A5	20q13	KCC2	K^+^, Cl^−^	NA	Neurones	Epilepsy [23]	Bumetanide, 55; Furosemide, 10; VU 0463271, 0.061 [24]; DIOA, ~10
SLC12A6	15q14	KCC3	K^+^, Cl^−^	Isoforms A and B [25]	Ubiquitous	Agenesis of the Corpus Callosum with Peripheral Neuropathy (ACCPN) [26]	Bumetanide, 40; Furosemide, 25; DIOA,~10
SLC12A7	5p15	KCC4	K^+^, Cl^−^	NA	Ubiquitous	Renal tubular acidosis [27]	Bumetanide, 900; Furosemide, 900; DIOA, ~10
SLC12A8	3q21	CCC9	NA	Six isoforms [28]	Ubiquitous	Psoriasis [28]	NA
SLC12A9	7q22	CIP1	NA	NA	Ubiquitous	NA	NA

NA, information not available.

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
