# Peer review of "Role of the Cation-Chloride-Cotransporters in Cardiovascular Disease"

_cells, 2020, doi:10.3390/cells9102293_

Round 1
Reviewer 1 Report
I read carefully this MS with a comprehensive review of the possible role of several cation/chloride co-transport proteins on several diseases. The physiological role of these proteins is more explored in relation to renal physiologgy. However, emerging evidences have show a putative role of these co-transporters to the physiological function of smooth muscle cells and neurons. The review emphasizes these aspects. I recommend publication considering it well writen and useful for readers.
Author Response
Many thanks for the comments, much appreciated.

Reviewer 2 Report
Please see attached file.

Author Response
Many thanks for the comments, much appreciated. See attached document, original comments are in red, and our reply is in bold, and the changes in the main text haven been highlighted in yellow.

Round 2
Reviewer 2 Report
The authors have for the most part addressed my concerns, and overall the manuscript is greatly improved. However, several concerns were not adequately addressed, some changes were not made, and there are some other minor errors.
Major
Page 1 line 30 – hypertension is not a cardiovascular disease per se, but is a major risk factor for CVD.
Page 1 line 36-42 “Hypertension is caused by water retention in the kidney, due to increased sodium reabsorption, which causes extracellular fluid (ECF) volume 37 expansion [4].” Not all hypertension is related to the kidney, so suggest changing to “The renal system plays a critical role in blood pressure homeostasis”.
Page 1 lines 40-42 – the authors have moved text about vascular phenotypes to this location, but it is still without context. The paragraph is describing cardiovascular disease, then suddenly jumps to vascular cells, without any rationale regarding its relevance to cardiovascular disease. My previous comment asked for clarification of the relevance of these changes to CVD, but this has not been done.
Page 2 lines 16-18 – it is not stated that KCCs are only weakly inhibited by NKCC blockers.
Figure 2 has not been corrected as stated. The issue regarding Apelin receptor activation leading to increased intracellular PDGF still exists. Membrane PDGF receptor is not shown.
Page 10 line 27 HCD has not been defined in the text as stated.
Page 10 lines 30-34. While the text has been modified, the statement that a bigger n and longer measurement period indicates conclusive evidence that NKCC1 plays a role in maintenance of vascular tone still remains. This must be corrected, since it is not a valid statement.
Page 10 lines 36-38. Ref 93 is a reference regarding use of telemetry as the gold standard, not a study with NKCC1 mice. Correct reference is https://pubmed.ncbi.nlm.nih.gov/18701622/ Both need to be cited.
Page 12 lines 7-9 – “Although mice that lack NKCC1 expression show impaired epithelial chloride secretion, later studies have not produced evidence to support a significant role of NKCC1 in chloride secretion [104].”
This should be changed to “Although mice that lack NKCC1 expression show impaired epithelial chloride secretion in gut and trachea (https://pubmed.ncbi.nlm.nih.gov/10480906/), later studies have not produced evidence to support a significant role of NKCC1 in chloride secretion in kidney https://pubmed.ncbi.nlm.nih.gov/11292635/.”
Abaci et al mentioned in the response is not related to NKCC1, but ref 104 is correct. Probably a copy-paste error.
Page 12 lines 42-43 “In the development of salt-sensitive hypertension (Dahl and DOCA), AVP secretion is dysregulated resulting in impairment in renal ionic absorption.” This needs references
Page 13 line 19. Ref 42 is not the correct reference. https://pubmed.ncbi.nlm.nih.gov/18955660/ should be used. Also change text to “ectopic expression of both full-length and ks-WNK1.”
Page 13 line 21 – the references is still incorrect and is from Uchida, not Lifton. Ref 42 should be used.
Page 13 line 47 - the word “recently” has not been removed as stated.
Page 13 line 7 - the authors have not addressed my previous comment correctly. Here is the comment:
Wouldn't the physiological effects of a SPAK inhibitor more resemble SPAK KO mice, which effects on the vasculature and kidney rather than cardiac cells? A major effect of SPAK knockout in mice is to mimic the effects of Gitelman syndrome/thiazide diuretics i.e. inhibition of NCC in the kidney leading to lower blood pressure. Ref 110 proposes a role in protection from stroke. Considering this, the disease relevance of an effect on cardiac cells is not clear. Is ref 110 accurate for cardiac cells considering reference is focused on brain? Is this speculation?
Authors’ response: The sentence on the effects of ZT-1a on cardiac cells is a speculation and the sentence have been reworded to clarify that further studies are needed.
The key point is that it is not clear why the authors are discussing Cl- homeostasis in cardiac cells, when a major effect of SPAK disruption is to alter vascular tone, which is more relevant to stroke (ref 6), and to alter NaCl reabsorption by the kidney. Discussion of cardiac cells is irrelevant.
Minor
Page 2 lines 11 and 12 “cotransporters” has extra space
Fig. 1 H-ATPase should be H+-ATPase
Page 10 line 27 – “NKCC1 could have hypotensive effects in vascular cells.” Should be changed to “NKCC1 inhibition could lower blood pressure through effects on vascular cells.”
Page 10 line 30, data values and n are still in text.
Page 14 line 4 should be “Forbush”
Page 14 line 16 “Bartter’s needs to be corrected to “Bartter”
Author Response
Thank you for reviewing our revised manuscript, "Role of the cation-chloride-cotransporters in cardiovascular disease" (Manuscript ID cells-921242). We have addressed all the comments as attached. Original comments are in red, and our reply is in bold, and the changes in the main text haven been highlighted in yellow.
